# Multilevel Regulation of Peroxisomal Proteome by Post-Translational Modifications

**DOI:** 10.3390/ijms20194881

**Published:** 2019-10-01

**Authors:** Luisa M. Sandalio, Cecilia Gotor, Luis C. Romero, Maria C. Romero-Puertas

**Affiliations:** 1Department of Biochemistry and Cellular and Molecular Biology of Plants, Estación Experimental del Zaidín, CSIC, 18008 Granada, Spain; maria.romero@eez.csic.es; 2Institute of Plant Biochemistry and Photosynthesis, CSIC and the University of Seville, 41092 Seville, Spain; gotor@ibvf.csic.es (C.G.); lromero@ibvf.csic.es (L.C.R.)

**Keywords:** nitric oxide, *S*-nitrosylation, nitration, peroxisome, persulfidation, phosphorylation, posttranslational modifications, reactive oxygen species, sulfenylation

## Abstract

Peroxisomes, which are ubiquitous organelles in all eukaryotes, are highly dynamic organelles that are essential for development and stress responses. Plant peroxisomes are involved in major metabolic pathways, such as fatty acid β-oxidation, photorespiration, ureide and polyamine metabolism, in the biosynthesis of jasmonic, indolacetic, and salicylic acid hormones, as well as in signaling molecules such as reactive oxygen and nitrogen species (ROS/RNS). Peroxisomes are involved in the perception of environmental changes, which is a complex process involving the regulation of gene expression and protein functionality by protein post-translational modifications (PTMs). Although there has been a growing interest in individual PTMs in peroxisomes over the last ten years, their role and cross-talk in the whole peroxisomal proteome remain unclear. This review provides up-to-date information on the function and crosstalk of the main peroxisomal PTMs. Analysis of whole peroxisomal proteomes shows that a very large number of peroxisomal proteins are targeted by multiple PTMs, which affect redox balance, photorespiration, the glyoxylate cycle, and lipid metabolism. This multilevel PTM regulation could boost the plasticity of peroxisomes and their capacity to regulate metabolism in response to environmental changes.

## 1. Introduction to Peroxisomal Metabolism, Function, and Dynamics

Peroxisomes, which are ubiquitous in all eukaryotes and essential for the development and stress responses of yeast, plants and animals, are organelles surrounded by a single membrane, with a very simple ultrastructure. They are involved in metabolic pathways, such as fatty acid β-oxidation, branched amino acid catabolism, ureide metabolism, reactive oxygen species (ROS), and reactive nitrogen species (RNS), some of which are shared cross-kingdoms [1,2]. However, other important specific metabolic functions such as photorespiration are exclusively associated with plant peroxisomes (Figure 1). A common characteristic of peroxisomes from different organisms is high ROS production which, in plants, is associated with metabolic pathways, such as fatty acid β-oxidation, ureide metabolism, sulfite oxidation, polyamine catabolism, and photorespiration. These organelles also have a great capacity to remove ROS thanks to a complex antioxidative defense systems such as catalase (CAT), superoxide dismutase (SOD), glutathione-*S*-transferase, and the ascorbate-glutathione cycle [1,3]. In addition, peroxisomes are an important source of RNS, thus it was reported to contain NO and GSNO, the latter acting as a cellular reservoir of NO [1,4].

Peroxisomal proteins are encoded in the nucleus and imported into the peroxisome throughout peroxisomal proteins called peroxines (PEXs). Peroxines PEX3, PEX6, and PEX19 participate in inserting peroxisomal membrane proteins (PMPs) into peroxisomal membrane (group II PMPs) or into a peroxisome-destined region of the ER membrane (group I PMPs) [2]. Peroxisomal matrix proteins are imported through soluble receptor PEX5, which recognizes proteins containing peroxisomal targeting signal 1 (PTS1) associated with the C terminal, and through soluble receptor PEX7 which recognizes peroxisomal targeting signal 2 (PTS2) associated with the N terminus, using a docking site composed of PEX13 and PEX14 [3]. Some matrix proteins do not contain PTSs and are imported into the peroxisome by piggyback [5]. The number of peroxisomes can be increased by fission during cell division or by proliferation, a process associated with stress conditions [6,7] which takes place through peroxisome elongation, constriction, and fission and is controlled by PEX11. This process is controlled by PEX11, which consists of five isoforms from PEX11a to PEX11e, dynamin-related proteins (DRPs), and fission proteins FISSION1 (FIS1) [6]. DRP3A and DRP3B regulate the division of peroxisomes and mitochondria, while DRP5 acts independently of DRP3 and is involved in the division of peroxisomes, chloroplasts, and mitochondria [8]. FIS1 has two homologs, FIS1A and FIS1B, which are shared by both peroxisomes and mitochondria [6].

Peroxisomes are highly dynamic organelles that are able to change their metabolic pathways, number, size, morphology, relationship with other organelles, and even speed of movement depending on development stage, cell type, tissue, and stress conditions [1,3,7]. Peroxisomes have a close relationship with other organelles. Oil bodies and peroxisomes collaborate in seeds and cotyledons to provide energy during the first stage of seedling germination and growth by channeling triacylglycerides from the oil bodies to peroxisomal β-oxidation, with Acyl CoA oxidase being one of the first enzymes involved in this process, which also produces H_2_O_2_ (Figure 1) [2,9,10]. Acetyl-CoA from fatty acid β-oxidation is then converted into C4 acids via the glyoxylate cycle, which is present in a type of peroxisome called glyoxysome (Figure 1). The C4 compounds are then transported to the mitochondria, where they can be either converted to malate and transported to the cytosol for gluconeogenesis, or can be used as substrates for respiration (Figure 1) [2,11]. The glyoxylate cycle is composed of the following enzymes: Isocitrate lyase (ICL) and malate synthase (MS), which are unique to this cycle, as well as malate dehydrogenase (MDH), citrate synthase (CS), and aconitase (AC) which are common to the glyoxylate cycle [11] (Figure 1) and the tricarboxylic acid (TCA) cycle in mitochondria [2]. Fatty acid β-oxidation is involved in several important processes including synthesis of indolacetic acid (IAA) from indolebutyric acid (IBA), jasmonic acid from 12-oxophytodienoic acid (OPDA) generated in the chloroplast (see [10] for review), ubiquinone, as well as secondary metabolites like benzoic acid (BA) and phenylpropanoids (Figure 1) [2,12,13].

Photomorphogenesis and senescence are examples of the metabolic plasticity of peroxisomes. During photomorphogenesis, light triggers a shift in the metabolism from fatty acid β-oxidation and the glyoxylate cycle to photorespiration or the oxidative photosynthetic carbon cycle, in which sugars are oxidized to CO_2_ under light in a complex pathway where several enzymes are distributed among chloroplasts, peroxisomes, and mitochondria requiring close physical contact between all these organelles [2] (Figure 1). One of the key peroxisomal enzymes in photorespiration is glycolate oxidase which oxidizes glycolate (produced in chloroplasts), giving rise to glyoxylate and H_2_O_2_. Glyoxylate is further transaminated to glycine by glutamate:glyoxylate aminotransferase, while glycine is transported to mitochondria to be converted to serine, producing CO_2_, ammonia and NADH (Figure 1). Serine is transported to peroxisomes where it is used to generate hydroxypiruvate by serine:glyoxylate aminotransferase. Hydroxypiruvate is, in turn, reduced to glycerate by hydroxypiruvate dehydrogenase and is transferred to the chloroplast where it is converted to 3-phosphoglycerate (Figure 1) [14]. The metabolic transition during photomorphogenesis requires the degradation of specific proteins, which become obsolete or of whole organelles, and imports of new proteins into peroxisomes to maintain photorespiration [2]. Peroxisomal protease LON2 plays an important role in the selective degradation of glyoxylate-dependent matrix proteins during this metabolic transition, although selective degradation of peroxisomes by pexophagy also helps to regulate the thiolase associated to fatty acid β-oxidation [15,16]. Both LON2 and pexophagy regulate peroxisome abundance [7,15], with LON2 chaperone activity acting as a pexophagy inhibitor [15]. During senescence, an opposite transition takes place through the down-regulation of photorespiration, while glyoxylate-associated proteins are enhanced [17].

Organelle communication is also an important issue with regard to signaling and metabolic network regulation. Peroxisomes are highly dynamic organelles whose speed of movement can change in response to disturbances in their environment which are regulated by ROS and Ca^++^ [7,18]. Peroxisomal extensions, also called peroxules, can be observed under oxidative stress conditions; although their precise function is unclear, it has been suggested that they are involved in peroxisome elongation prior to division [19] and also in the mechanism that regulates rapid cell responses to redox changes and signaling [7]. PEX11a has been reported to be involved in peroxule formation triggered by ROS-dependent NADPH oxidases [7]. However, the redox-sensitive proteins involved in peroxisomal protein import and proliferation in plants have yet to be identified, which would help us to understand the signaling network that governs rapid responses to environmental changes and the role of ROS/RNS in this process. In yeast and human fibroblasts, PEX5 has been reported to act as a redox switch that regulates the import of peroxisomal matrix proteins into peroxisomes [20,21], although it has not, so far, been described in plants. In addition, peroxisome population control can be regulated by pexophagy at the basal level [22] and under stress conditions [23,24], which could be crucial for rapid cell responses.

Peroxisomes, which share metabolites with other organelles and are involved in the biosynthesis of different hormones (jasmonic acid, indolacetic acid and salicylic acid) and signals (ROS, RNS) and in the perception of rapid responses to stress, play an important role in metabolism and transcriptome regulation [1]. The perception of environmental changes, as well as acclimation and adaptive responses to these changes in plants, are complex processes which involve different signal transduction pathways acting in a coordinated way to finally regulate gene expression and protein functionality through post-translational modifications, giving rise to elaborate, specific rapid responses. Although gene transcription modulation governs protein content, gene expression does not always coincide with protein activity [4]. This is due to protein post-translational modifications (PTMs) that can regulate protein activity, localization, inter-protein interactions and degradation, giving rise to rapid finely tuned regulation of protein function and consequently of metabolic pathways and signaling processes [25]. Over 200 different PTMs have been reported in mammals and plant cells [26], including carbonylation, *S*-nitrosylation, nitration, phosphorylation, ubiquitination, persulfidation, glutathionylation and SUMOylation, whose functionality can differ depending on the protein. PTM regulation of proteins becomes increasingly complex due to the synergistic or antagonistic interplay between the different PTMs. In this review, we provide up-to-date information on putative PTMs of peroxisomal proteins, as well as a framework to further investigate the function and crosstalk of PTMs in peroxisomes, with a particular focus on those associated with ROS and NO.

## 2. Post-Translational Modification of Peroxisomal Proteins

To analyze putative PTMs capable of modulating peroxisomal proteins, we compared a list of peroxisomal candidate proteins from *Arabidopsis thaliana*, which undergo PTMs, from the following databases: Plant PTM Viewer [27]; TAIR [28] with the aid of GO codes for peroxisomes and glyoxysomes (GO: 0005777 and GO: 0009514, respectively); and NCBI [29] with a list of peroxisomal proteins [30] using the Venny tool [31]. Putative peroxisomal proteins from these databases were selected from proteomic studies available in the literature [30] and taking into account putative PTSs sequences [28]. Approximately 35% of the putative peroxisomal proteins were identified as PTM targets. To provide an overview of the putative role of peroxisomal protein PTMs, the proteins were classified into the principal peroxisomal and cellular pathways. However, further research is required to demonstrate that most of these proteins are modified by the PTMs described in this review and to determine their function in regulating metabolism and signaling events.

### 2.1. H_2_O_2_-Dependent Post-Translational Modifications

#### 2.1.1. Protein Sulfenylation

Oxidative protein modifications, such as sulfenylation, sulfinylation and sulfonylation, as well as intra- and inter-molecular disulfide bond formation, are rapid and reversible mechanisms (except irreversible sulfonylation) to regulate protein function in living cells in response to changing redox states [32]. Sulfur-containing amino acids such as cysteine are highly susceptible to oxidation by ROS such as H_2_O_2_. This reactive oxygen species leads to reversible oxidation of cysteines to SOH which is highly reactive and can be overoxidized to sulfinic acid (SO_2_H) and sulfonic acid (SO_3_H) [33] under excessive ROS accumulation conditions. Electrophilic SOH also reacts with a free thiol on the same protein or other proteins, giving rise to the formation of intramolecular and intermolecular disulfides, and can even react with GSH-promoting S-glutathionylated disulphides [33]. SOH also forms sulfenylamide through the reaction of sulfur with the backbone nitrogen of the adjacent residue. Due to their transient nature, these sulfur modifications, which can be reversibly reduced by thioredoxin and glutaredoxin pathways, are regarded as redox switches (for review see [33]). For example, OxyR, the H_2_O_2_-sensitive transcription factor in *E. coli*, becomes active when oxidized by H_2_O_2_ and inactive when reduced [33]. An example from peroxisomal proteome is 3-ketoacyl-CoA thyolase [34]. As reported with regard to *Arabidopsis* cytosolic malate dehydrogenase 1, sulfenylated cysteines can mediate disulfide formation to prevent protein overoxidation and are also involved in the protein catalytic site and therefore in regulating the function of this protein as well as in the case of peroxisomal dehydroascorbate reductase 2 (DHAR2) [35]. Thus, the effect of this PTM on proteins differs depending on the protein involved, giving rise to reversible or irreversible inactivation by overoxidation [33]. However, the role of sulfenylation in protein degradation due to protein overoxidation is not fully understood [36].

Cysteine sulfenylation has been reported to be crucial for redox signaling, with several sulfenylated proteins suggested to be oxidative stress sensors [37]. The identification of cellular sulfenylated proteins and sulfenome could therefore be very useful in detecting potential redox sensors of ROS signaling pathways. Mass spectrometry-based techniques and proteomics have been used to identify sulfenylated proteins in plants (for review see [33]). A biotin switch method originally designed to detect protein *S*-nitrosylation has been adapted by changing the reducing agent from ascorbate to arsenite in order to identify sulfenylated proteins (for review see [32]). Western blot analysis using an anti-cysteine sulfenic acid antibody also enables sulfenylated proteins to be identified [36]. To image in vivo sulfenylated proteins, 5,5-Dimethyl-1,3-cyclohexanedione (dimedone), containing a nucleophilic carbon that selectively reacts with electrophilic sulfenic acids, as well as other more permeable analogs (DAz-1, DAz-2 and DYn-2), have been used (reviewed by [33]). A genetically-encoded probe based on YAP1 from yeast has been successfully used to analyze sulfenomes in *Arabidopsis* cell cultures [32,35].

##### Sulfenylated Peroxisomal Proteins

Using different proteomic analytical techniques, the following *Arabidopsis* peroxisomal proteins have been identified: Monodehydroascorbate reductase 1 and 4, found to be targets of Trx in *Arabidopsis* roots [36], and glutathione-disulfide reductase (GR). Both GR and MDHAR are important components of the ascorbate-glutathione cycle involved in antioxidant defenses. The glyoxalase 1 (GLX1) homolog is another putative sulfenylated protein involved in methylglyoxal detoxification and protection against carbonyls [38]. Other sulfenylated proteins are Acyl-CoA oxidase 1 (ACX1) and peroxisomal 3-ketoacyl-CoA thiolase 3 (AtKAT3), both involved in fatty acid β-oxidation. AtKAT2 is inactive in the oxidized dimer form and active in the reduced monomer form; depending on the peroxisomal redox environment, AtKAT2 controls β-oxidation and metabolite channeling by regulating the formation of a complex containing thiolase and multifunctional protein 2 (MFP2) [34]. The protein phosphatase 2A 55 kDa regulatory subunit B alpha isoform from *Medicago* has also been identified as a target of sulfenylation [36]. However, the effect of sulfenylation on proteins such as ACX and GLX has not been fully explained.

#### 2.1.2. Protein Carbonylation

Oxidative damage to proteins can be caused by excessive ROS-induced oxidation, giving rise to a variety of covalent protein modifications such as carbonylation of specific amino acids. The most frequently studied oxidation-dependent PTM is protein carbonylation which mainly targets proline, lysine, arginine, and threonine [39]. Carbonylation is caused by nucleophilic attacks on the CO group by OH radicals produced in Fenton-type reactions involving H_2_O_2_, O_2_^−^ and metals (mainly Fe and Cu [40]). Oxidation of proline or arginine side chains forms glutamic semialdehyde, while lysine oxidation results in aminoadipic semialdehyde, thus introducing carbonyl groups (R–C=O) into the protein structure [41]. The hydroxyl group of threonine side chains can also be oxidized to form the carbonyl group [41]. Protein carbonylation is regarded as an oxidative stress marker which gives rise to the inactivation and further degradation of proteins [42,43]. Although carbonylation is an irreversible protein modification, protein decarbonylation has been reported in mammals [41]. Several carbonylated proteins, previously characterized in mammals, yeast, and bacteria, have also been found to be oxidized in plants. This suggests that protein carbonylation could be more than a random process and be involved in controlling common biological functions in living organisms [44]. Protein carbonylation has been reported at different stages of the plant cycle [44]. Proteins damaged by carbonylation are preferentially degraded by proteasome 20S [45] but can also be removed by autophagy [46].

Protein carbonylation can be analyzed using a method based on derivatization with 2,4-dinitrophenylhydrazine (DNPH) which reacts with the carbonyl group of aldehyde or ketone and forms a hydrazone derivative (DNP), detectable by Western blot with the aid of specific antibodies against DNP [42,47] and by HPLC [48,49].

##### Carbonylated Peroxisomal Proteins

As mentioned above, peroxisomes, which are a major cellular source of ROS, are vulnerable to oxidation mainly under stress conditions when ROS break through antioxidant defenses. Under control conditions, peroxisomal proteins show a basal level of carbonylation which is increased by stress induced by Cd treatment [42], Cu plus ascorbate [50] and salinity [48]. Although the Plant PTM Viewer does not identify any putative peroxisomal targets of carbonylation, by using DNPH derivatization and HPLC several carbonyl target proteins have been identified in *Arabidopsis* plants, mainly associated with nucleic acid metabolism (nucleoside diphosphate kinase, NDPK), antioxidant defenses (CAT), and fatty acid β-oxidation (AIM1) [48]. The carbonylated proteins detected in response to Cd in pea plants include CAT, GR, and Mn-SOD, while MS, IL and MDH have been reported in castor bean endosperm exposed to Cu and ascorbate [50]. The activity of the target protein is considerably reduced by this type of protein modification [42,50], while carbonylated proteins have been found to be efficiently recognized and degraded by peroxisome proteases [42]. It has been suggested that inactive oxidized CAT is involved in recognizing oxidized peroxisomes to be targeted for pexophagy under basal and stress conditions, as phagophores and autophagy markers are located close to the peroxisomal core containing inactive catalase [22,23,51]. However, the mechanism involved remains unknown, as the peroxisomal matrix enzyme CAT is unable to interact directly with ATG8 or the receptor NBR1.

### 2.2. NO-Dependent PTMs

Nitric oxide (NO), a highly reactive gaseous free radical, plays an important regulatory role. Over the last twenty years, NO has been involved in plant processes ranging from development [52,53] and defence responses to both biotic and abiotic stresses [54,55,56]. Up to now, NO’s principal mode of action in plants has been reported to be non-classical signalling, which depends on covalent protein post-translational modifications. These PTMs are carried out by a sequence of reactive nitrogen species (RNS) resulting from the reaction of NO with other signalling molecules such as free radicals. The best known NO-dependent PTM in plants is *S*-nitrosylation, also called S-nitrosation (for review see [57] and [58]) which involves the formation of a nitrosothiol in a Cys residue and modifies the function, location, and stability of a number of proteins [59,60]. *S*-nitrosylation facilitates gene regulation through the modification of transcription factors (TFs) [61,62] and the DNA methylation index by altering enzymes involved in the methylation cycle [63]. NO interacts with most phytohormone-dependent signalling pathways through the *S*-nitrosylation of enzymes involved in biosynthetic pathways and/or phytohormone-dependent regulatory proteins [64,65,66] and regulates ROS levels through the *S*-nitrosylation of ROS-producing and -scavenging enzymes [67]. In plants, *S*-nitrosylation may result in the establishment of a disulfide bond, which is regarded as an intermediate path leading to a more stable modification [68]. In large-scale proteomic studies, over 1000 proteins have been shown to be putative targets of *S*-nitrosylation in plants [69,70], although the functional impact of this PTM has only been analyzed in approximately 2% of these proteins [60,70,71].

Less is known about the NO-dependent PTM tyrosine nitration, associated with nitrogen dioxide (NO_2_) and peroxynitrite (ONOO^−^), whose impact on protein functionality in plants has also been described [72]. Certain gaps in our knowledge of this PTM, such as its reversibility in plants, remain. ONOO^−^ can also induce the production of oxygenated forms of Cys residue, such as sulfenic acid (–SOH), sulfinic acid (–SO_2_H) and sulfonic acid (–SO_3_H), and *S*-glutathionylation [59]. Fewer putative targets of nitration have been identified in plants as compared to those of *S*-nitrosylation [73,74,75]. On the other hand, the best known direct modification of NO in animal systems, though less studied in plants, is the formation of complex bonds with transition metal ions in heme groups, such as guanylate cyclase, Cyt p450, and haemoglobin [76], with the haemoglobin heme group having also been described in plants [77,78,79,80].

#### Peroxisomal Targets of NO-Dependent PTMs

NO has been shown to be present in plant peroxisomes under physiological and stress conditions [81,82,83]. Several studies, mostly based on the use of NOS inhibitors, have suggested the presence of NOS-like activity in plants, which has also been described in peroxisomes [83], although no orthologue to the animal NOS-like gene has been found in land plants [84]. Peroxisomes also contain xanthine oxidase (XOD), which is capable of generating NO under special conditions [85]. The NO derivatives, peroxinitrite (ONOO–), and nitrosoglutathione (GSNO), have also been observed in peroxisomes [4,86], whose formation is promoted by NO in the presence of O_2_^.−^ and GSH, respectively. The protein targets of NO-dependent PTMs are therefore expected to be present in these organelles. Unless specifically located or associated with a particular gene, it has been difficult to pinpoint the location of the proteins obtained in the organelles using large-scale proteomic analyses. The peroxisomal targets of NO-dependent PTMs were obtained from large-scale *S*-nitrosylation and nitration proteomic analysis in *Arabidopsis* plants and compared with the in house peroxisomal protein list [71,74,87,88,89] and are described in Table 1 and Appendix A. Approximately 8.4% of the putative peroxisomal proteins are considered to be targets of *S*-nitrosylation (Appendix A). However, only 1.6% are regarded as putative targets of Tyr-nitration (Appendix A), suggesting that more specific studies, with purified organelles, may be required to determine the actual number of nitration targets. Peroxisomal putative *S*-nitrosylated proteins are involved in antioxidant defences (DHAR, GR, and CAT), photorespiration (alanine-2-oxoglutarate aminotransferase 2, HPR, alanine: glyoxylate aminotransferase, glutamate: glyoxylate aminotransferase), the glyoxylate cycle (isocitrate dehydrogenase, NAD-malate dehydrogenase 1 and 2, and citrate synthase 2); β-oxidation (ACX2,3 and 6, acyl-CoA synthase, 3-ketoacyl-CoA thiolase 3 and 4), the pentose phosphate cycle (6-phosphogluconate dehydrogenase family protein), the serine/threonine protein phosphatase 2A and nucleoside diphosphate kinase (Table 1 and Appendix A).

A proteomic study of peroxisomes isolated from pea plants shows that six *S*-nitrosylation target proteins are involved in photorespiration (HPR, GOX, serine:glyoxylate aminotransferase and aminotransferase 1), β-oxidation (MDH) and the antioxidant system (CAT) [4] (Table 1). Furthermore, the *S*-nitrosylation levels of both H_2_O_2_-producing GOX and H_2_O_2_-removing CAT change under abiotic stresses such as cadmium and 2,4-D, suggesting that *S*-nitrosylation plays a role in regulating peroxisomal H_2_O_2_ under physiological and stress conditions by reducing the activity of these proteins [4]. Nitrated proteins, which are mainly involved in ROS metabolism (CAT, MDHAR, Cu, Zn-SOD), photorespiration (MDH, GOX) and β-oxidation (MDH), have also been identified in peroxisomes [72,73,74,90,91] (Table 1).

### 2.3. H_2_S-Dependent Post-Translational Modifications: Protein Persulfidation

The perception of hydrogen sulfide (H_2_S) as a toxic molecule has been superseded by an extraordinary number of studies which highlight its important role in plant physiology. H_2_S, which is involved in regulating various processes essential for plant performance, has been established to be a signaling molecule equal in importance to NO and H_2_O_2_ in plant systems [92,93].

H_2_S plays a role in plant stress responses which enables plants to adapt to adverse conditions and positively affects seed germination, root elongation, and overall plant survival. Studies have been carried out on its involvement in drought, salinity, hypoxia, heat, as well as chilling and metal stress, while (pre) treatment with H_2_S and endogenous sulfide induction have mostly been found to alleviate oxidative stress by enhancing antioxidant plant defenses [94,95].

The vital plant processes, in which hydrogen sulfide plays a signaling role, are photosynthesis [96], programmed cell death [97], and autophagy [98,99], while the number of regulated processes is continuously increasing. Another important aspect of H_2_S is the regulation of stomatal movement which has major consequences for efficient plant water regulation. Various studies have shown that hydrogen sulfide plays an active role in abscisic acid-dependent signaling pathways in guard cells by regulating ion channel activity and interacts in a complex way with other signaling molecules and hormones [100,101,102,103,104].

Although much information is available on the different pathways and processes regulated by H_2_S, little is known about the precise mechanism of action involved. The mechanism underlying how hydrogen sulfide functions is related to its chemical properties and high level of reactivity [95,105]. H_2_S shows a high affinity for metalloprotein metal, and sulfide inhibits the mitochondrial electron transport chain by covalently binding to the iron center of cytochrome c oxidase. Its nucleophilic properties also suggest that H_2_S can act as an oxidative stress reductant by reacting with reactive oxygen species (ROS), which has been suggested as how sulfide treatment increases plant cell antioxidant capacity. Finally, sulfide is also involved in the PTM, now called persulfidation (formerly known as *S*-sulfhydration), which oxidizes cysteine thiol groups to persulfide groups (R-S-SH).

The first proteomic analyses of persulfidation in plants were carried out on *Arabidopsis* leaf tissues grown in soil under long-day photoperiod conditions using two different experimental methods based on specific biotin labelling of persulfide groups [106]. These analyses revealed that persulfidation is a widespread PTM in the plant proteome which is involved in a wide range of biological functions and in regulating important processes [107].

#### Persulfidated Peroxisomal Proteins

The proteomic analyses carried out in wild type *Arabidopsis* plants grown under controlled conditions identified more than 3000 proteins [107], 61 of which were assigned subcellular locations in peroxisomes (Table 1 and Appendix A). These include the following enzymes identified in the antioxidant system CAT3 and Cu-Zn superoxide dismutase 3 (SOD3); fatty acid oxidation enzymes such as ACX1, 3, 4, and 6 and two enoyl-CoA hydratase/isomerase enzymes (AIM1/ECHIA). Sulfite oxidase (SOX), which catalyzes two-electron oxidation of sulfite to sulfate derived from sulfur amino acid catabolism, was also identified. 

Persulfidation can affect the subcellular localization of a protein and/or its enzymatic properties [108,109]. Although in vitro studies have shown that treatment with NaHS inhibits catalase activity in a dose-dependent manner, they do not specify whether this inhibition is due to cysteine persulfidation or to a sulfide reaction with the heme center [110]. Similar results have also been reported with regard to NADP-dependent isocitrate dehydrogenase and the NADP-malic enzyme from sweet pepper, in which NaHS treatment results in partial enzymatic inhibition [111,112].

The specific chemical reactions causing protein persulfidation depend on the environment [105]. This modification requires an oxidant, as hydrogen sulfide cannot directly modify cysteine residues to form a persulfide group. So, the ionic form of sulfide, the thiolate (HS−) can react with cysteine sulfenic acid (R-SOH) to form cysteine persulfide (R-SSH). Partially oxidized sulfide species such as polysulfides (H_2_Sn) are able to transfer sulfane sulfur (S0) atoms to reduced cysteine to form persulfides or polysulfided cysteine residues (R-SSnH). The oxidative environment of peroxisomes may therefore be appropriate to form these oxidized intermediaries to enable persulfidation to take place. Despite the capacity of peroxisomes to generate different reactive oxygen and nitrogen species that can easily react with H_2_S, the reduced H_2_S molecule has recently been detected in peroxisomes using a specific fluorescent probe [110].

### 2.4. Phosphorylation

Protein phosphorylation, which is a critical step in many signal transduction pathways and a reversible process regulated by kinases and phosphatases, is one of the most important and probably best known protein modification in both prokaryotes and eukaryotes. Phosphorylation mainly takes place on serine, tyrosine, and threonine, with serine being the most common site (85%) [113]. Protein kinases transfer the phosphate group from ATP to the hydroxyl group of Ser, Thr, and Tyr residues, while protein phosphatases hydrolyze the phosphoester bond to dephosphorylate proteins [114]. Changes in protein phosphorylation are one of the most common mechanisms used by the cell to regulate the activity, functionality, subcellular location, inter-protein interactions, and turnover of different proteins [103,114]. Over recent years, there has been a considerable growth in studies describing different ways of analyzing phosphoproteins: immobilized metal affinity chromatography (IMAC), titanium dioxide metal oxide affinity chromatography (TiO2-MOAC) coupled with mass spectrometric analysis and immunoprecipitation (IP) of tyrosine phosphorylated proteins and peptides with high affinity antiphosphotyrosine antibodies [115]. Thanks to these techniques, a wide-range of proteins have been identified as targets of phosphorylation in several plant tissues and species. 

#### Phosporylated Peroxisomal Proteins 

Phosphoproteomes, kinases, and phosphatases present in *Arabidopsis* peroxisomes have recently been extensively reviewed [116]. Kataya et al. [116] have gathered information on validated peroxisomal and peroxisome-related proteins, as well as on meta-analysis of phosphoproteins available in the literature to compile a list of almost 100 phosphorylated peroxisomal proteins. Based on these data and those available in Plant PTM Viewer [27], we have reviewed a list of putative peroxisomal phosphoproteins (Table 1 and Appendix A). The following kinases have been reported to be associated with peroxisomes: Protein kinases 1, 2, 3, 5, and 6 (PK1,PK2,PK3,PK5,PK6) [117], a protein kinase present in glyoxysomes (GPK1, also called protein kinase 7 (PK7) [118], calcium-dependent protein kinase 1 (CDPK1) [119], and protein kinase constitutive triple response 1 (CTR1), which regulates ethylene signaling suppression [116,120]. Peroxisomal phosphatases include protein phosphatase 2A regulatory (B) subunit ‘θ (PP2A-B’θ), purple acid phosphatase 7 (PAR 7), POL-like phosphatase 2 (PLL2), protein phosphatase 2A catalytic (C) subunit 2 (PP2A-C2), protein phosphatase 2A catalytic (C) subunit 5 (PP2A-C5), and protein phosphatase 2A scaffolding (A) subunit 2 (PP2A-A2) (reviewed in [116]).

Protein phosphorylation is involved in the regulation of cross-talk between peroxisomes and other organelles such as in the case of photorespiration (GOX, SAGT1, GGT1, and HPR1) [121] (Table 1 and Appendix A). GOX has been reported to be phosphorylated in pea plants exposed to the herbicide 2,4-D [122]. Several enzymes, including ACX4 and ACX6, MFP2, malonyl-CoA decarboxylase family protein, enoyl-CoA hydratase/isomerase, and acetyl-CoA synthetase, involved in different steps in the fatty acid β-oxidation pathway, are also phosphorylated. The enzymes citrate synthase 1, 2, and 3, and NAD-malate dehydrogenase 2 are involved in the glyoxylate cycle, while glutathione peroxidase 2, MDHAR 1 and 4, CAT, and SOD are involved in antioxidative defences [116] (Table 1 and Appendix A). An increase in SOD3 phosphorylation was observed in pea leaves exposed to the herbicide 2,4-D [122]. Some studies of *Arabidopsis* plants show that CAT3 is phosphorylated by CPK8 [123] and that CAT3 and CAT2 are phosphorylated by SOS2, a class 3 sucrose-nonfermenting 1-related kinase, and by nucleoside diphosphate kinase 2 (NDPK2), with CAT being activated by phosphorylation mainly under stress conditions [124]. However, as CPK8, SOS2, and NDPK2 are not associated with peroxisomes, CAT may play a role in the cytosol [124]. It has not yet been established that any of the peroxisomal kinases identified play a role in CAT regulation in these organelles, although NDPK1 present in peroxisomes (Appendix A) could be involved. In mammalian cells, the tyrosine phosphorylation of CAT is critical for ubiquitination-dependent degradation of this protein [125].

Certain components of the pentose phosphate cycle and NADP-isocitrate dehydrohenase are also found in phosphorylared peroxisomal proteins [116], as well as several proteins involved in amino acids and ureide metabolism, peroxisome biogenesis (several PEXs), division/proliferation (PEX11 and DRP), and peroxisomal protein degradation (LON2) [116] (Table 1 and Appendix A).

### 2.5. Other post-Translational Protein Modifications

Ubiquitylation is involved in targeting misfolded or unnecessary proteins for proteasome degradation and plays an essential role in cellular responses to stress conditions and in regulating phytohormone signaling [126]. More than 30 peroxisomal proteins to be ubiquitylated have been identified, which include: NAD(P)H dehydrogenases, several kinases, heat shock proteins, zinc finger-related proteins, cytoskeleton-related proteins, β-oxidation- and glyoxalate cycle-dependent proteins (citrate synthase 3), and antioxidants (APX and CAT) (Appendix A). However, the most studied ubiquitinated peroxisomal protein is peroxin5 (PEX5) in *S. cerevisiae* [127], which is involved in importing peroxisomal proteins through peroxisome biogenesis and in exporting proteins to the cytosol for degradation by the proteasome [128,129]. In *S. cerevisiae,* PEX5 monoubiquitination enables a peroxisome-tethered ATPase complex to recycle PEX5 to the cytosol for further rounds of cargo recruitment, while PEX5 polyubiquitination by the cytosolic UBC4, together with the peroxisomal ubiquitin-protein ligase PEX2, targets PEX5 for proteasomal degradation [127]. Although PEX5 ubiquitination has not yet been demonstrated in plants, mutants defective in peroxisome-associated ubiquitination machinery (*pex2-1*, *pex4*, *pex10-2*, and *pex12-1*) are involved in plant peroxisomal import and PEX5 retrotranslocation [2].

Protein acetylation is characterized by the attachment of an acetyl group either to the N-terminus of the protein (irreversible modification) or to lysine residues (reversible modification). This modification is dynamically regulated by acyltransferases and deacetylases, although non-enzymatic acetylation by acetyl-CoA and NAD has also been reported [130]. An N-terminal acetyltransferase (AtNatA) has been reported in *Arabidopsis*, containing catalytic (NAA10) and auxiliary (NAA15) subunits, which is involved in embryogenesis, endosperm development, plant immunity and responses to drought [131]. Nevertheless, our understanding of the physiological effects of acetylation, as well as the overall interplay and regulation of the acetylome itself in plants, remains elusive [132]. In mammalian cells, Nt-acetylation determines the subcellular localization of certain proteins and modulates inter-protein interactions and protein folding [132]. Table 1 and Appendix A show putative acetylated peroxisomal proteins including several antioxidants, proteins involved in Ca^2+^ homeostasis, fatty acid β-oxidation, photorespiration, the glyoxylate cycle, NAD(P)H recycling, protein degradation, phosphorylation, and folding, as well as those involved in peroxisome biogenesis and proliferation, stress responses, and hormone biosynthesis. However, these data require experimental confirmation.

Asparagine (N)-linked glycosylation (ALG) of proteins, one of the most common PTMs in plants, is involved in protein folding and stability, secretion and interactions with ligands and other proteins [133]. Thirteen N-glycosylated peroxisomal proteins have been identified in this work (Table 1 and Appendix A), some of which are involved in fatty acid β-oxidation, protein degradation, stress responses and photorespiration. 

Peroxisomes contain a large number of N-terminal proteolytic target proteins. These are involved in uric metabolism, fatty acid β-oxidation, phosphorylation, the glyoxylate cycle, ubiquitination, protein degradation, antioxidant defenses, stress responses, peroxisome biogenesis and proliferation, sulfur metabolism, NAD(P)H recycling, photorespiration, and protein degradation (Table 1 and Appendix A).

## 3. Crosstalk between PTMs in the Regulation of Peroxisomal Metabolism 

In biological systems, proteins are frequently modified by various PTM events [26]. Computational analysis of proteomes from different organisms has shown that proteins undergoing PTMs engage in more interactions and are positioned in more central locations than non-PTM proteins [26]. Analysis of whole peroxisomal proteomes shows that a very large number of peroxisomal proteins are targeted by multiple PTMs (Table 1 and Appendix A), which affect redox balance, the ASC-GSH cycle, photorespiration, the glyoxylate cycle, and secondary lipid metabolism (Figure 2, Table 1 and Appendix A). PTMs overlap on the same protein more extensively than expected, a pattern commonly observed in different species [26], thus demonstrating the importance of multilevel PTM regulation. This boosts the plasticity of peroxisomes and their capacity to regulate metabolism in response to changes in the environment. 

A computational analysis by Duan and Walther [26] revealed that, of the twelve types of PTM studied in mammals, yeast, and *Arabidopsis*, phosphorylated proteins were most consistently situated in central network locations and most affected by other PTMs. This means that phosphorylation of a single protein potentially modulates many different interactions and molecular processes simultaneously [26]. In plants, protein phosphorylation is an important mechanism involved in regulating β-oxidation to supply energy and carbon metabolites and to activate the peroxisomal part of the JA biosynthesis pathway (Figure 1; extensively reviewed by [116]). Several key enzymes involved in IAA and SA biosynthetic pathways, which share certain β-oxidation steps, are known phospho-proteins (Figure 1) [116].

ACX, a key β-oxidation enzyme, is an important source of H_2_O_2_ in peroxisomes. Six isoenzymes (ACX1-6), which differ with regard to substrate and play distinct overlapping roles in fatty acid oxidation, are present in *Arabidopsis* [134]. As shown in Table 1, ACX isoforms also differ in relation to their regulation by PTMs; ACX1 is targeted by the largest number of PTMs (*S*-nitrosylation, nitration, methionine oxidation, phosphorylation, persulfidation, Nt-proteolysis and L-acetylation), while ACX2 is only targeted by *S*-nitrosylation. However, the effect of each PTM on ACX activity has not been established. Recently, CAT2 was reported to physically interact with and increase ACX3 and ACX4 activity in vitro [135]. This interaction may protect ACX enzymes against oxidative H_2_O_2_ degradation and therefore regulate JA biosynthesis, which is essential for plant pathogen defense responses [135]. SA, in turn, suppresses CAT2, leading to an increase in H_2_O_2_ and sulfenylation of tryptophan (Trp) synthetase b subunit 1 (TSB1) located in the chloroplast, thus limiting this precursor of auxin synthesis [135]. CAT also interacts with GOX [136], probably as a defense mechanism against oxidative damage. 

Although less is known about the effect of *S*-nitrosylation on β-oxidation in *Arabidopsis*, protein *S*-nitrosylation mediated by *S*-nitroso-CoA (SNO-CoA) in yeast and mammalian cells is associated with a specific SNO-CoA reductase encoded by the alcohol dehydrogenase 6 protein [137], which protects acetoacetyl–CoA thiolase against inhibition by *S*-nitrosylation [137]. However, no presence of SNO-CoA reductase has been reported in plant peroxisomes.

With respect to the interplay between different PTMs, some sulfenylated proteins identified in *Arabidopsis* are *S*-glutathionylated, others contain redox-active disulfide bonds, while another group is *S*-nitrosylated [36]. *S*-glutathionylation (S-SG) is a ubiquitous redox-sensitive reversible cysteine modification which protects against overoxidation and regulates protein activity [138], while no *S*-glutathionylated peroxisomal proteins have been identified. Interplay between carbonylation, *S*-nitrosylation, and Tyr-nitration has been reported in citrus plants exposed to salinity stress. Under these conditions, *S*-nitrosylation prevents ROS and RNS oxidative damage to several proteins involved in the Calvin–Benson cycle, probably by inducing conformational changes in specific proteins [139]. Cross-talk between NO, H_2_S, ABA, and polyamines is involved in acclimation processes in citrus plants (reviewed in [140]). Another example of protection against carbonylation by *S*-nitrosylation is C4 phosphoenolpyruvate carboxylase in sorgum under salinity stress conditions [141]. Some studies also corroborate the protective role played by *S*-nitrosylation in mammalian cells under oxidative stress conditions, which prevents protein carbonylation [72,142].

Other studies demonstrate an antagonistic interplay between protein Tyr nitration and phosphorylation in different biological systems including plants [72,143]. Both these PTMs compete for the same Tyr sites and can therefore interfere with different cellular processes, such as microtubule organization and cell signaling, via MAP kinase cascades [72], although these interactions have not been explored in peroxisomal proteomes. Interestingly, most phosphorylated peroxisomal proteins involved in β-oxidation do not show nitration, although ACXs and acyl-CoA synthase are *S*-nitrosylated (Table 1 and Appendix A). In mammalian cells, *S*-nitrosylation inhibits phosphorylation-based processes such as signal transduction by inducing conformational changes in kinases or by blocking their autophosphorylation, which affect kinase substrate specificity and selectively modify different protein isoforms [144].

Photorespiration is an important metabolic pathway which facilitates the recycling of 2-phosphoglycolate carbon atoms produced by ribulose-1,5-bisphosphate carboxylase/oxygenase (Rubisco) in the chloroplast, as well as the removal of potentially toxic metabolites. Although different PTMs, shown in Table 1 and Appendix A, could modulate most enzymes involved in photorespiration, the regulation of this cycle is little understood. Protein phosphorylation, which commonly occurs in photorespiration enzymes [121], appears to be a highly complex process due to the possible interplay of different PTMs such as *S*-nitrosylation, nitration, persulfidation, and acetylation (Table 1). The contribution of each PTM to the regulation of this pathway depends on the redox environment inside the peroxisome. CAT, SOD, DHAR, MDHAR, APX, and GPX, the main enzymes involved in antioxidant defenses in peroxisomes, are also targets of different PTMs (Table 1), which facilitate finely tuned regulation of ROS levels to regulate redox signaling and protection of other proteins against oxidative damage.

Peroxisomal dynamics can be regulated by multiple PTMs. Thus, as mentioned previously in yeast and human fibroblasts, PEX5 redox changes regulate peroxisomal biogenesis by controlling the import of peroxisomal matrix proteins [2], although this has not been demonstrated in plants. As shown in Appendix A, most PEXs are targeted only by phosphorylation, except PEX14, which is targeted acetylation and N-terminus proteolysis, and pPEX19 which is also *S*-nitrosylated. Interestingly, PEX11 phosphorylation is an important issue in *P pastoris* for peroxisome division [145]. Organelle degradation by autophagy in yeast and mammalian cells is regulated by multiple, mutually exclusive PTMs, such as acetylation, phosphorylation and ubiquitilation, which target the same lysine and thus provide opportunities for cross-regulation [146]. Although not fully understood in plants, the mechanism involved in the regulation of pexophagy by PTMs is triggered by redox changes and protein carbonylation [22,23,51]. While several PEXs have been reported to be phosphorylated in *Arabidopsis* (Appendix A), a direct relationship between ROS, PEX, phosphorylation, and pexophagy has not been established. In the methylotrophic yeast *Hansenula polymorpha*, phosphorylated PEX14p proteins are needed for pexophagy, while nonphosphorylated proteins are involved in protein import [147]. Peroxisomal quality control could also be regulated by the PTM cross-talk of proteases such as LON2 which can be regulated by N-terminus proteolysis, N-acetylation, and phosphorylation (Appendix A). The formation of peroxisomal extensions (peroxules) is regulated by ROS and PEX11a [7], although no redox-dependent PTMs of this protein have been reported. However, as previously mentioned, PEX11a, PEX11c, and PEX11b are identified in this review as targets of phosphorylation, while PEX11d is also acetylated and nitrated, suggesting that these PTMs on PEX11 may play an important role in peroxisomal biogenesis and inter-organelle communication.

## 4. Conclusions

Peroxisomes, which are essential for plant development and stress responses, are characterized by considerable metabolic plasticity and act as a redox metabolism and signaling hub. This plasticity depends on a complex network of PTMs which regulate essential metabolic pathways, hormone balance, and inter-organelle cross-talk. However, further functional studies of the role played by most of the PTMs studied in each protein’s activity and degradation, and in the interplay and hierarchy of the different PTMs in peroxisomes, are required.

## Figures and Tables

**Figure 1 ijms-20-04881-f001:**
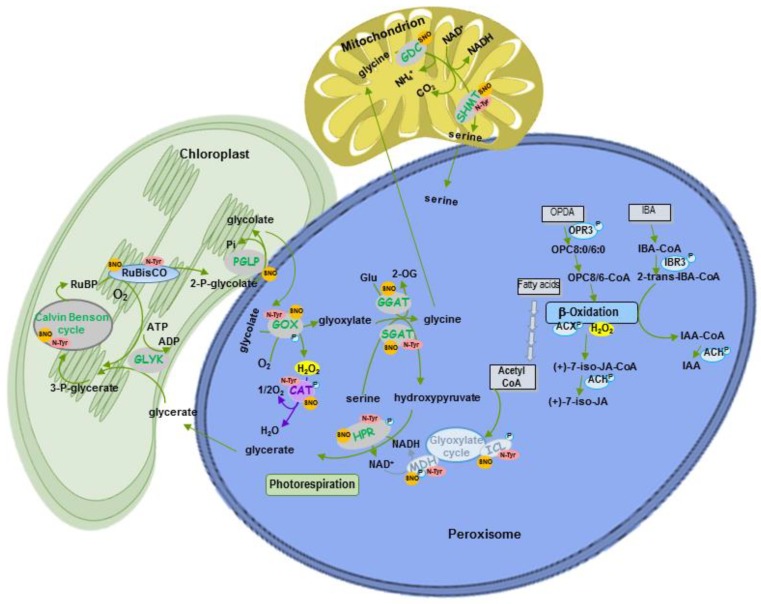
Representative scheme of the main peroxisomal metabolic pathways and their regulation by NO-dependent protein post-translational modifications (PTMs) and phosphorylation. Ph, phosphorylation; SNO, *S*-nitrosylation; N-Tyr, nitration.

**Figure 2 ijms-20-04881-f002:**
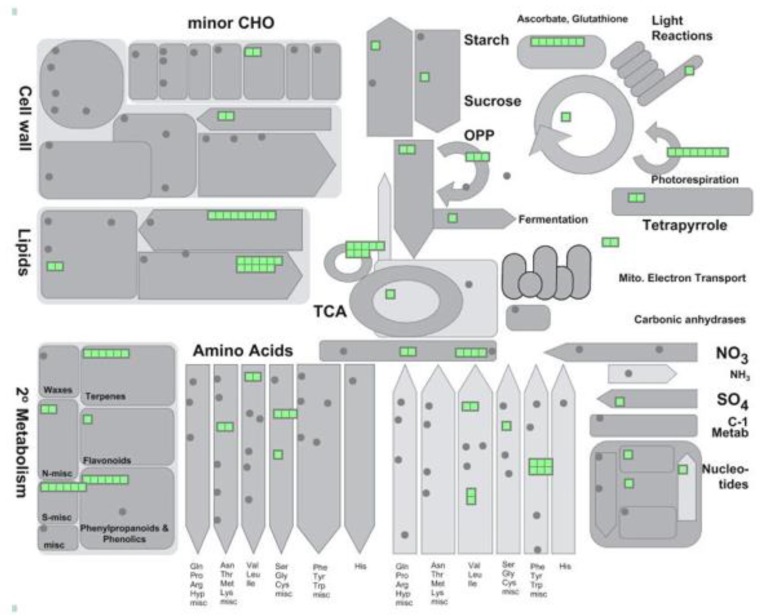
Putative perxisomal protein targets of different PTMs (Table 1 and Appendix A) were classified and visualized in metablic pathways using MapMan software (metablism overview; https://mapman.gabipd.org/).

**Table 1 ijms-20-04881-t001:** Peroxisomal processes and proteins regulated by multiple PTMs.

Locus	Description	PTM_Types
**REDOX/ASC-GLU CYCLE**		
AT1G19570	dehydroascorbate reductase	ac; na; no; nt; ro
AT2G31570	glutathione peroxidase 2	ac; na; nt; ph
AT3G24170	glutathione-disulfide reductase	ac; mo; na; no; nt; sf
AT3G27820	monodehydroascorbate reductase 4	ph; sf
AT3G52880	monodehydroascorbate reductase 1	ac; mo; ph; ps
AT4G35000	ascorbate peroxidase 3	ac; na; nt; ro; ub
AT4G35970	ascorbate peroxidase 5	na; nt; ro
AT1G20620	catalase 3	ac; na; no; nt; ph; ps; ntyr
AT1G20630	catalase 1	ac; no; nt; ph
AT4G35090	catalase 2	ac; na; no; nt; ph; ps; ca
AT5G18100	copper/zinc superoxide dismutase 3	ph
**PHOTORESPIRATION**		
AT1G68010	hydroxypyruvate reductase	ac; mo; na; no; nt; ph; ps
AT1G70580	alanine-2-oxoglutarate aminotransferase 2	ac; na; no; nt; ph; ro; ps
AT2G13360	alanine:glyoxylate aminotransferase	ac; na; no; nt; ph; ps; ntyr
AT3G14130	Aldolase-type TIM barrel family protein/ glycolate oxidase	ac; na
AT3G14150	Aldolase-type TIM barrel family protein/ glycolate oxidase	na
AT3G14415	Aldolase-type TIM barrel family protein/ glycolate oxidase	ac; na; ng; nt; ph; ntyr
AT3G14420	Aldolase-type TIM barrel family protein/ glycolate oxidase	ac; na; ng; nt; ph; ntyr
AT4G18360	Aldolase-type TIM barrel family protein/ glycolate oxidase 3	ac; na; ph
AT4G39660	alanine:glyoxylate aminotransferase 2	ac
AT1G23310	glutamate:glyoxylate aminotransferase	ac; na; no; nt; ph; ro; ub; ps
**GLUCONEO/GLYOXYLATE CYCLE**		
AT1G54340	isocitrate dehydrogenase	na; no; ro; ps
AT2G22780	peroxisomal NAD-malate dehydrogenase 1	no; nt; ps
AT2G42790	citrate synthase 3	ac; nt; ph; ub; ps
AT3G16910	acyl-activating enzyme 7	ac; nt; ps
AT3G21720	isocitrate lyase	na; nt
AT3G58740	citrate synthase 1	ac; nt; ph
AT3G58750	citrate synthase 2	ac; no; nt; ph; ps
AT5G03860	malate synthase	na; nt
AT5G09660	peroxisomal NAD-malate dehydrogenase 2	ac; mo; no; nt; ph; ro; ps; ntyr
**LIPID METABOLISM-β OXIDATION**		
AT4G16760	acyl-CoA oxidase 1	ac; mo; na; nt; ps; sf
AT5G65110	acyl-CoA oxidase 2	no
AT1G06290	acyl-CoA oxidase 3	no; nt; ps
AT1G06310	acyl-CoA oxidase 6	no; ph; ps
AT1G20560	acyl activating enzyme 1	na; nt
AT1G66120	AMP-dependent synthetase and ligase family protein	ub
AT1G77540	Acyl-CoA N-acyltransferases (NAT) superfamily protein	na; nt; ph
AT1G76150	enoyl-CoA hydratase 2	na; nt; ro; ps
AT1G60550	enoyl-CoA hydratase/isomerase D	ph; ps
AT2G30200	EMBRYO DEFECTIVE 3147	ac; mo; nt; ph
AT3G05970	long-chain acyl-CoA synthetase 6	nt; ps
AT3G06810	acyl-CoA dehydrogenase-like protein	ac; na; ph; ps
AT3G06860	multifunctional protein 2	ac; nt; ph; ps
AT3G51840	acyl-CoA oxidase 4	na; nt; ph; ps
AT4G04320	malonyl-CoA decarboxylase family protein	ph
AT4G16210	enoyl-CoA hydratase/isomerase A	ac; mo; na; nt; ph; ps
AT4G27780	acyl-CoA binding protein 2	ph
AT4G29010	Enoyl-CoA hydratase/isomerase family	ac; mo; na; nt; ps; ca
AT5G16370	acyl activating enzyme 5	na; nt
AT5G27600	long-chain acyl-CoA synthetase 7	nt
AT5G36880	acetyl-CoA synthetase	ac; na; no; nt; ph
AT5G42890	sterol carrier protein 2	mo; na; nt
AT4G00520	Acyl-CoA thioesterase family protein	ph
AT5G48880	peroxisomal 3-keto-acyl-CoA thiolase 2	ac; nt; ph; ps
AT2G33150	peroxisomal 3-ketoacyl-CoA thiolase 3	ac; mo; no; nt; ub; ps; sf
AT4G14440	3-hydroxyacyl-CoA dehydratase 1	ph
AT3G26820	Esterase/lipase/thioesterase family	ph

Carbonylation (ca), Lysine Acetylation (ac), Lysine Ubiquitination (ub), Methionine Oxidation (mo), N-glycosylation (ng), N-terminal Acetylation (na), N-terminus Proteolysis (nt), N-terminal Ubiquitination (nu), Phosphorylation (ph), Reversible Cysteine Oxidation (ro), S-Nitrosylation (no), persulfidation (ps), sulfenylation (sf), nitration (ntyr).

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
