# Peer review of "Multilevel Regulation of Peroxisomal Proteome by Post-Translational Modifications"

_ijms, 2019, doi:10.3390/ijms20194881_

Round 1

Reviewer 1 Report

In this review paper, Sandalio et al. provide a thorough perspective on post-translation modifications that target peroxisomal proteins in plants. Also, the authors have gathered a variety of data that is scattered over several databases and compiled them in this manuscript. The conclusion is that at least some of these PTMs affect the activities of some proteins. The text reads well, and is useful for researchers working on PTMs and plants. I have some suggestions, all are minor.

Line 40, “peroxiredoxines” should be “peroxiredoxins”

Line 54, “DRPS” and “FISS”. The plural of DRP should be “DRPs” (s lower case). There are several proteins involved in fission: The one the authors are referring to is, I believe, FIS1.

Line 59, After the statement that “Peroxisomes have a close relationship with other organelles”, a full stop would sound better.

Lines 62-63, The sentence “Acetyl-CoA is then converted into sucrose…, for further gluconeogenesis and mitochondrial respiration” is not clear. Sucrose should be succinate.

Line 77, “glycolate” should be glyoxylate?

Line 88, “thyolase” should be “thiolase”

Lines 124-125, “Sulfonylation”. I had the impression that this modification is not reversible. Please verify.

Lines 198-199, “Inactive oxidized CAT is also…”. The sentence as written gives the idea that the authors are referring to intraperoxisomal CAT. If so, how is this protein recognized by the cytosolic autophagy machinery? Please clarify.

Section “3- Protein persulfidation”, 4th paragraph. “H2S shows a high affinity metalloprotein metal”. It should be “shows a high affinity for metalloprotein…”

Section “Persulfidated peroxisomal enzymes”. First paragraph. “isomeraseenzymes”. Please introduce a space between the two words.

Section “5- other posttranslational protein modifications”. The last line of this section refers to cysteine oxidation which was already addressed in previous sections. Is this correct?

Author Response

Response to Reviewer 1

All changes in the manuscript are written in red

Line 40, “peroxiredoxines” should be “peroxiredoxins”

This has been corrected

Line 54, “DRPS” and “FISS”. The plural of DRP should be “DRPs” (s lower case). There are several proteins involved in fission: The one the authors are referring to is, I believe, FIS1.

The mistakes have been corrected and more information about DRPs and FIS1 has been included.

Line 59, After the statement that “Peroxisomes have a close relationship with other organelles”, a full stop would sound better.

This has been corrected

Lines 62-63, The sentence “Acetyl-CoA is then converted into sucrose…, for further gluconeogenesis and mitochondrial respiration” is not clear. Sucrose should be succinate.

We have included more information to better explain the pathways involved.

Line 77, “glycolate” should be glyoxylate?

This has been corrected

Line 88, “thyolase” should be “thiolase”

This has been corrected

Lines 124-125, “Sulfonylation”. I had the impression that this modification is not reversible. Please verify.

This has been corrected

Lines 198-199, “Inactive oxidized CAT is also…”. The sentence as written gives the idea that the authors are referring to intraperoxisomal CAT. If so, how is this protein recognized by the cytosolic autophagy machinery? Please clarify.

More information has been included.

Section “3- Protein persulfidation”, 4th paragraph. “H2S shows a high affinity metalloprotein metal”. It should be “shows a high affinity for metalloprotein…”

This has been corrected

Section “Persulfidated peroxisomal enzymes”. First paragraph. “isomeraseenzymes”. Please introduce a space between the two words.

This has been corrected

Section “5- other posttranslational protein modifications”. The last line of this section refers to cysteine oxidation which was already addressed in previous sections. Is this correct?

Yes, the paragraph has been removed.

Reviewer 2 Report

This review article summarizes post-translational modifications found in the proteome of plant peroxisome proteins. Much emphasis is put on the descriptions of ROS- or RNS- mediated modifications, such as sulfenylation and nitrosylation, in consideration of multiple ROS-generating reactions that the organelle contains. The discussion mentions plausible cross-talks of the protein modifications, which highlights the physiological significance of the modifications.

The logical structure of this manuscript is well organized, yet in several points the descriptions leaves us vague impressions due to lacks of concrete protein names. Below I ‘d like to mention each point that can be improved, including minor grammatical errors.

Page 2 (line 78): The phrase ‘or even whole organelles’ should be ‘or of even whole organelles’. Page 3 (line 99): The term pexophagy is already introduced in line 81, and so ‘selective peroxisome autophagy, called pexophagy’ should be shortened to ‘pexophagy’. Page 3 (line 129): For the designations of SO2H and SO3H, the numbers should be subscript. Page 4 (line 134 and 135): The authors introduce reversible sulfur modifications toward certain proteins as redox switches. The concrete protein names should be mentioned, since this information holds great importance in this manuscript. Page 6 (line 247 to 248): Here the authors show that 35% of the peroxisome proteins are found modified. Is this frequency significantly higher than the percentages for the rest of the cellular components, or that for the total cellular proteome Page 9 (line number missing, the 20th line from the top): Maybe the phrase ‘high affinity to metalloprotein metal’ is correct. Page 11 (line number missing, the 17th line from the top): The concrete mutant names of peroxisome-associated ubiquitination machinery should be mentioned here. Page 11 (the paragraph describing Protein acetylation): Here the authors don’t mention anything about acetyltransferases. Briefly introduce the identified acetyltransferases in plants, if any.

Author Response

Response to Reviewer 2

All changes in the manuscript are written in red

Page 2 (line 78): The phrase ‘or even whole organelles’ should be ‘or of even whole organelles’.

This has been corrected.

Page 3 (line 99): The term pexophagy is already introduced in line 81, and so ‘selective peroxisome autophagy, called pexophagy’ should be shortened to ‘pexophagy’.

This has been corrected.

Page 3 (line 129): For the designations of SO2H and SO3H, the numbers should be subscript.

This has been corrected.

Page 4 (line 134 and 135): The authors introduce reversible sulfur modifications toward certain proteins as redox switches. The concrete protein names should be mentioned, since this information holds great importance in this manuscript.

Some examples of redox switche proteins have been included in the manuscript.

 Page 6 (line 247 to 248): Here the authors show that 35% of the peroxisome proteins are found modified. Is this frequency significantly higher than the percentages for the rest of the cellular components, or that for the total cellular proteome

We were unable to  determine   the percentage  of PTMs in chloroplasts and mitochondria in order to compare it with the results observed in this study of peroxisomes.

Page 9 (line number missing, the 20th line from the top): Maybe the phrase ‘high affinity to metalloprotein metal’ is correct.

This has been corrected.

 Page 11 (line number missing, the 17th line from the top): The concrete mutant names of peroxisome-associated ubiquitination machinery should be mentioned here.

This information has been included.

Page 11 (the paragraph describing Protein acetylation): Here the authors don’t mention anything about acetyltransferases. Briefly introduce the identified acetyltransferases in plants, if any.

The information required has been included in the text.

Reviewer 3 Report

The review authored by Dr. Sandalio and colleagues provides an extensive up-to-date overview of the most common redox- and non-redox-related posttranslational modifications (PTMs) of proteins
involved in the biology of plant peroxisomes. In addition, where possible, the authors also discuss
the functional implications of these PTMs. In general, the manuscript is certainly of interest to
IJMS readers and provides a useful framework to researchers in the field. However, before being accepted for publication, the authors should address the following points:
1. A major concern is that, within their manuscript, the authors claim multiple times to provide an up-to-date review on post-translational modifications of peroxisomal proteins. However, upon careful analysis, this claim is somewhat misleading. Indeed, several of the proteins discussed in the manuscript are no bona fide peroxisomal proteins. The underlying reason is that the peroxisomal localization of many of the proteins that have been identified in peroxisome-enriched fractions in large-scale proteomics studies have not been
experimentally verified, and accepting such information unthinkingly may yield
physiologically non-relevant conclusions. For example, to the best knowledge of this reviewer, peroxisomes do neither contain a photosystem nor DNA or a DNA transcription machinery, and listing proteins such as PSBP-2, chloroplast RNA binding protein, histone H4, and others in a Table with "putative peroxisomal targets" is – from a scientific point of view – not correct. The authors, who are experts in the field, should present all their data in a
coherent, concise, and critical way. To cope with this criticism, the authors should indicate in each Table which of the proteins listed are bona fide peroxisomal proteins, non-peroxisomal proteins involved peroxisome functioning, or potential contaminants (e.g, in case the largescale
proteomics data have never been experimentally validated). In case the authors are fully convinced that their list of “peroxisomal” proteins is correct, they should provide the reader with more background information regarding the potential physiological role of photosystem proteins, ribosomal proteins, and DNA- and RNA-binding proteins within Peroxisomes.
2. The structure of the manuscript is not logic. On one hand, the title of Section 1 is “Redoxdependent post-translational modifications”. On the other hand, two redox-related PTMs (in casu “NO-dependent PTMs” and “protein persulfidation”) are discussed under different headings. This is confusing.
3. Page 1, line 44: “Peroxisomal proteins are encoded in the nucleus and imported into the peroxisome throughout peroxisomal membrane proteins (PMPs) called peroxines (PEXs).”  This statement is not correct: peroxins are proteins involved in peroxisome biogenesis, and for people outside the field – it is confusing to state that peroxins are PMPs. Some PMPs (e.g., PEX13 AND PEX14) are indeed peroxins, but other PMPs are not; and some peroxins (e.g., PEX5 and PEX7) are involved in the import of matrix proteins across the peroxisomal membrane, but these proteins are no PMPs. Please change the text accordingly.
4. Multiple sentences read awkward or are, from a scientific point of view, not correct (given that the page and line numbers are not properly formatted, it is difficult to pinpoint the location of these sentences in an appropriate manner).
PMPs are inserted into peroxisomes with the aid of PEX3, PEX6 and PEX19 directly into peroxisomal membrane proteins (group II PMPs) or into a peroxisome-destined region of the ER membrane [2].
Peroxisomes have a close relationship with other organelles, oil bodies and
peroxisomes collaborate in seeds and cotyledons to provide energy during the first stage of seedling germination and growth by channeling fatty acids to peroxisomal β-oxidation, with acyl CoA oxidase being one of the first enzymes to produce H2O2 (Fig.1) [2,8,9].
This metabolic transition requires the degradation of specific proteins, which become obsolete or even whole organelles, and imports of new proteins into peroxisomes to maintain photorespiration [2].
During senescence, an opposite transition takes place through down-photorespiration, while glyoxylate-associated proteins are enhanced [17].
“Peroxisomes, which share metabolites with other organelles and are involved in the biosnthesis of different hormones (jasmonic acid, indolacetic acid and salicylic acid) and signals (ROS, RNS and Ca++) and in the perception of fast responses to stress, ….”
® Peroxisomes are not involved in the BIOSYNTHESIS of Ca++.
“As mentioned above, peroxisomal proteins, which are a major cellular source of ROS, are vulnerable to oxidation”. Given that some peroxisomal proteins (e.g., catalase), can also degrade ROS, this statement is misleading.
“Inactive oxidized CAT is also recognized by autophagy machinery to trigger pexophagy under basal and stress conditions [15, 22, 45].” This statement yields the impression that CAT physically interacts with components of the autophagy machinery (e.g., NBR1, ATG8a). However, from a topological point of view, this is impossible.
“Furthermore, the S-nitrosylation levels of both H2O2-producing (GOX) and H2O2-removing CAT change under abiotic stresses such as cadmium and 2,4-D, suggest that S-nitrosylation plays a role in regulating peroxisomal H2O2 levels under physiological and stress conditions [4].”
“These include the following enzymes identified in the antioxidant system involved in hydrogen peroxide metabolism: CAT3 and Cu-Zn superoxide dismutase 3 (SOD3); fatty acid oxidation enzymes such as ACX1, 3, 4 and 6 and two enoyl-CoA hydratase/isomeraseenzymes (AIM1/ECHIA).”
“Phosphorylation mainly takes place on serine, tyrosine and threonine, with serine being the most common site (85%) for phosphorylation [109].” Delete “for phosphorylation”.
“CPK8 can specifically phosphorylate CAT3 and regulate its activity as well as H2O2 homeostasis [119].” Given that CPK8 is localized to the plasma membrane, also this statements needs further explanation for the reader.
“Reversible cysteine oxidation is another important PTM in peroxisomes (Table S1).”
Why is this statement listed in Section 5, and not in Section 1?
5. The authors should rigorously check the manuscript’s errors in spelling and grammar. Once again, given that the page and line numbers are not properly formatted, it is difficult to clearly describe all these syntactic and grammatical inaccuracies in a concise manner.
peroxisines → peroxins
“… ureide metabolism and polyamine metabolism, in the biosynthesis of jasmonic, …”
→ “… ureide and polyamine metabolism, the biosynthesis of jasmonic, …”
“protein post-transcriptional modifications (PTMs)” → ”protein post-translational
modifications (PTMs).
“… targeted by multiple PTMs, which affects redox balance” → “… targeted by multiple PTMs, which affect redox balance”
“(see for review 9,),” → “(for review, see [9])”
“thyolase” → “thiolase”
“In yeast and human fibroblasts, PEX5 has been reported to act as a redox switch that regulates the import of peroxisomal matrix proteins into peroxisomes [20-21], although …”. Reference 21 does not discuss the redox switch properties of human PEX5. Please refer instead to doi: 10.1111/tra.12129 (or doi:
10.1016/j.bbamcr.2017.07.013).
Sulfenilated → sulfenylated
“(TFs; [55, 56]]” → “(TFs; [55, 56])”
“… although no orthologous to animal NOS-like gene has been found in land plants” →
“… although no orthologue to animal NOS-like gene has been found in land plants”
“(reviewed in [112].” → “(reviewed in [112]).”
“However, in mammal cells the …” → “However, in mammalian cells, the …”
“phtotorespiration” → “photorespiration”
“A computational analysis by Duan and Walther [25] reveals that” → “A computational
analysis by Duan and Walther [25] revealed that”

Author Response

Response to reviewer 3

All changes in the manuscript are written in red

A major concern is that, within their manuscript, the authors claim multiple times to provide an up-to-date review on post-translational modifications of peroxisomal proteins. However, upon careful analysis, this claim is somewhat misleading. Indeed, several of the proteins discussed in the manuscript are no bona fide peroxisomal proteins. The underlying reason is that the peroxisomal localization of many of the proteins that have been identified in peroxisome-enriched fractions in large-scale proteomics studies have not been experimentally verified, and accepting such information unthinkingly may yield physiologically non-relevant conclusions. For example, to the best knowledge of this reviewer, peroxisomes do neither contain a photosystem nor DNA or a DNA transcription machinery, and listing proteins such as PSBP-2, chloroplast RNA binding protein, histone H4, and others in a Table with "putative peroxisomal targets" is – from a scientific point of view – not correct. The authors, who are experts in the field, should present all their data in a coherent, concise, and critical way. To cope with this criticism, the authors should indicate in each Table which of the proteins listed are bona fide peroxisomal proteins, non-peroxisomal proteins involved peroxisome functioning, or potential contaminants (e.g, in case the largescale proteomics data have never been experimentally validated). In case the authors are fully convinced that their list of “peroxisomal” proteins is correct, they should provide the reader with more background information regarding the potential physiological role of photosystem proteins, ribosomal proteins, and DNA- and RNA-binding proteins within Peroxisomes.

We agree with the reviewer and, in order to avoid confusion, we have removed  those  proteins  not been clearly associated with peroxisomes from both tables.

The structure of the manuscript is not logic. On one hand, the title of Section 1 is “Redoxdependent post-translational modifications”. On the other hand, two redox-related PTMs (in casu “NO-dependent PTMs” and “protein persulfidation”) are discussed under different headings. This is confusing.

We have changed  Redox-dependent to H2O2-dependent in the title and also  the term Persulfidation to  H2S-dependent post-translational modifications.

Page 1, line 44: “Peroxisomal proteins are encoded in the nucleus and imported into the peroxisome throughout peroxisomal membrane proteins (PMPs) called peroxines (PEXs).” This statement is not correct: peroxins are proteins involved in peroxisome biogenesis, and for people outside the field – it is confusing to state that peroxins are PMPs. Some PMPs (e.g., PEX13 AND PEX14) are indeed peroxins, but other PMPs are not; and some peroxins (e.g., PEX5 and PEX7) are involved in the import of matrix proteins across the peroxisomal membrane, but these proteins are no PMPs. Please change the text accordingly.

We apologize for the mistakes and have corrected the text according with the reviewer’s suggestion.

Multiple sentences read awkward or are, from a scientific point of view, not correct (given that the page and line numbers are not properly formatted, it is difficult to pinpoint the location of these sentences in an appropriate manner).

PMPs are inserted into peroxisomes with the aid of PEX3, PEX6 and PEX19 directly into peroxisomal membrane proteins (group II PMPs) or into a peroxisome-destined region of the ER membrane [2].

This sentence has been changed to make the meaning clearer.

Peroxisomes have a close relationship with other organelles, oil bodies and peroxisomes collaborate in seeds and cotyledons to provide energy during the first stage of seedling germination and growth by channeling fatty acids to peroxisomal β-oxidation, with acyl CoA oxidase being one of the first enzymes to produce H2O2 (Fig.1) [2,8,9].

The sentence has been changed.

This metabolic transition requires the degradation of specific proteins, which become obsolete or even whole organelles, and imports of new proteins into peroxisomes to maintain photorespiration [2].

The sentence has  been changed.

During senescence, an opposite transition takes place through down-photorespiration, while glyoxylate-associated proteins are enhanced [17].

The sentence has been changed.

“Peroxisomes, which share metabolites with other organelles and are involved in the biosnthesis of different hormones (jasmonic acid, indolacetic acid and salicylic acid) and signals (ROS, RNS and Ca++) and in the perception of fast responses to stress, ….”

® Peroxisomes are not involved in the BIOSYNTHESIS of Ca++.

We apologise for the mistake and have removed Ca  from the sentence.

“As mentioned above, peroxisomal proteins, which are a major cellular source of ROS, are vulnerable to oxidation”. Given that some peroxisomal proteins (e.g., catalase), can also degrade ROS, this statement is misleading.

As mentioned in the sentence, this  refers to the  conditions under which  ROS overproduction  takes place.

“Inactive oxidized CAT is also recognized by autophagy machinery to trigger pexophagy under basal and stress conditions [15, 22, 45].” This statement yields the impression that CAT physically interacts with components of the autophagy machinery (e.g., NBR1, ATG8a). However, from a topological point of view, this is impossible.

We have included a sentence to  explain  this  more clearly.

“Furthermore, the S-nitrosylation levels of both H2O2-producing (GOX) and H2O2-removing CAT change under abiotic stresses such as cadmium and 2,4-D, suggest that S-nitrosylation plays a role in regulating peroxisomal H2O2 levels under physiological and stress conditions [4].”

This sentence has been changed.

“These include the following enzymes identified in the antioxidant system involved in hydrogen peroxide metabolism: CAT3 and Cu-Zn superoxide dismutase 3 (SOD3); fatty acid oxidation enzymes such as ACX1, 3, 4 and 6 and two enoyl-CoA hydratase/isomeraseenzymes (AIM1/ECHIA).”

This sentence has been slightly changed.

 “Phosphorylation mainly takes place on serine, tyrosine and threonine, with serine being the most common site (85%) for phosphorylation [109].” Delete “for phosphorylation”.

This has been corrected.

“CPK8 can specifically phosphorylate CAT3 and regulate its activity as well as H2O2 homeostasis [119].” Given that CPK8 is localized to the plasma membrane, also this statements needs further explanation for the reader.

We have included a sentence to explain the phosphorylation of CAT3 and CAT2 in the cytosol.

“Reversible cysteine oxidation is another important PTM in peroxisomes (Table S1).”Why is this statement listed in Section 5, and not in Section 1?

This sentence has been removed, as  the information is already given  for  H2O2 dependent PTMs.

The authors should rigorously check the manuscript’s errors in spelling and grammar. Once again, given that the page and line numbers are not properly formatted, it is difficult to clearly describe all these syntactic and grammatical inaccuracies in a concise manner.

peroxisines → peroxins

“… ureide metabolism and polyamine metabolism, in the biosynthesis of jasmonic, …”

→ “… ureide and polyamine metabolism, the biosynthesis of jasmonic, …”

“protein post-transcriptional modifications (PTMs)” → ”protein post-translational

modifications (PTMs).

“… targeted by multiple PTMs, which affects redox balance” → “… targeted by multiple PTMs, which affect redox balance”

“(see for review 9,),” → “(for review, see [9])”

“thyolase” → “thiolase”

All these spelling and grammatical mistakes have been rectified, and the manuscript has been corrected by  native English speakers.

“In yeast and human fibroblasts, PEX5 has been reported to act as a redox switch that regulates the import of peroxisomal matrix proteins into peroxisomes [20-21], although …”. Reference 21 does not discuss the redox switch properties of human PEX5. Please refer instead to doi: 10.1111/tra.12129 (or doi:10.1016/j.bbamcr.2017.07.013).

As suggested by the reviewer, we have changed the reference.

Sulfenilated → sulfenylated

“(TFs; [55, 56]]” → “(TFs; [55, 56])”

“… although no orthologous to animal NOS-like gene has been found in land plants” →

“… although no orthologue to animal NOS-like gene has been found in land plants”

“(reviewed in [112].” → “(reviewed in [112]).”

“However, in mammal cells the …” → “However, in mammalian cells, the …”

“phtotorespiration” → “photorespiration”

“A computational analysis by Duan and Walther [25] reveals that” → “A computational

analysis by Duan and Walther [25] revealed that”

All these mistakes have been corrected.

Round 2

Reviewer 3 Report

The authors have satisfactorily addressed the majority of my concerns. However, despite the fact that the manuscript has been proofread by a native English speaker, some moderate English changes are still required during the proofreading process.